# Identification of *APTX* disease-causing mutation in two unrelated Jordanian families with cerebellar ataxia and sensitivity to DNA damaging agents

Nidaa A. Ababneh[1]*, Dema Ali[1⦵], Ban Al-Kurdi[1⦵], Malik Sallam[2,3,4], Abdulla M. Alzibdeh[5], Bareqa Salah[6], Abdee T. Ryalat[7], Belal Azab[2,8], Basil Sharrack[9], Abdalla Awidi[1,10,11]*

**1** Cell Therapy Center, The University of Jordan, Amman, Jordan, **2** Department of Pathology, Microbiology and Forensic Medicine, School of Medicine, The University of Jordan, Amman, Jordan, **3** Department of Clinical Laboratories and Forensic Medicine, Jordan University Hospital, Amman, Jordan, **4** Department of Translational Medicine, Faculty of Medicine, Lund University, Malmö, Sweden, **5** School of Medicine, the University of Jordan, Amman, Jordan, **6** General Surgery Department/Plastic & Reconstructive, Jordan University Hospital, The University of Jordan, Amman, Jordan, **7** King Hussein Cancer Center, Amman, Jordan, **8** Department of Human and Molecular Genetics, School of Medicine, Virginia Commonwealth University, Richmond, Virginia, United States of America, **9** Academic Department of Neuroscience and Sheffield NIHR Neuroscience BRC, Royal Hallamshire Hospital and The University of Sheffield, Sheffield, United Kingdom, **10** Hemostasis and Thrombosis Laboratory, School of Medicine, the University of Jordan, Amman, Jordan, **11** Department of Hematology and Oncology, Jordan University Hospital, Amman, Jordan

⦵ These authors contributed equally to this work.
* nidaaanwar@gmail.com, n.ababneh@ju.edu.jo (NAA); abdalla.awidi@gmail.com (AA)

## Abstract

### Background

Ataxia with oculomotor apraxia type 1 (AOA1) is a rare autosomal recessive cerebellar ataxia, caused by mutations in the *APTX* gene. The disease is characterized by early-onset cerebellar ataxia, oculomotor apraxia and severe axonal polyneuropathy. The aim of this study was to detect the disease-causing variants in two unrelated consanguineous Jordanian families with cerebellar ataxia using whole exome sequencing (WES), and to correlate the identified mutation(s) with the clinical and cellular phenotypes.

### Methods

WES was performed in three affected individuals and segregation analysis of p.W279* *APTX* candidate variant was performed. Expression levels of *APTX* were measured in patients' skin fibroblasts and peripheral blood mononuclear cells, followed by western blot analysis in skin fibroblasts. Genotoxicity assay was performed to detect the sensitivity of *APTX* mutated cells to $H_2O_2$, MMC, MMS and etoposide.

### Results

A recurrent homozygous nonsense variant in *APTX* gene (c.837G>A, p.W279*) was revealed in all affected individuals. qRT-PCR showed normal *APTX* levels in peripheral

**Data Availability Statement:** All relevant data are within the paper and its Supporting Information files.

**Funding:** This work was supported by the deanship of scientific research, University of Jordan, grant (2018- 2017/2). The funding organization did not play a role in the study design, data collection and analysis, decision to publish, or preparation of the manuscript and only provided financial support in the form of research materials.

**Competing interests:** The authors have read the journal's policy and have the following competing interests: author Belal Azab is currently affiliated with Prevention Genetics. However, they were affiliated with the University of Jordan at the time of the study. There are no patents, products in development or marketed products associated with this research to declare. This does not alter our adherence to PLOS ONE policies on sharing data and materials.

blood and lower levels in fibroblast cells. However, western blot showed the absence of *APTX* protein in patients' skin fibroblasts. Significant hypersensitivity to $H_2O_2$, MMC and etoposide and lack of sensitivity to MMS were noted.

## Conclusions

This is the first study to report the identification of a nonsense variant in the *APTX* gene (c.837G>A; p.W279*) in AOA1 patients within the Jordanian population. This study confirmed the need of WES to assist in the diagnosis of cerebellar ataxia and it emphasizes the importance of studying the pathophysiology of the *APTX* gene.

## 1. Introduction

Autosomal recessive cerebellar ataxias (ARCAs) are clinically and genetically heterogonous group of disorders [1]. Of these ARCAs, ataxia with oculomotor apraxia type 1 (AOA1) is a disease characterised by early-onset cerebellar ataxia, oculomotor apraxia and severe sensorimotor peripheral axonal polyneuropathy [2]. The genetic cause of AOA1 is a mutation in the *APTX* gene on chromosome 9p13, which consists of seven exons and encoding for aprataxin (APTX) nuclear protein [3]. Aprataxin is a member of the histidine triad superfamily, which consists of three major domains: forkhead-associated, histidine, and zinc finger domains [4]. Aprataxin is considered a component of DNA end-processing, which is involved in the removal of adenosine monophosphate (AMP) from the 5'-termini of DNA breaks [5, 6]. Following the abortive attempts of DNA ligase to join the complementary strands during DNA replication, APTX helps creating a gap for single-stranded DNA break (SSB) repair. This suggests a potential role of APTX as a "proofreader" during DNA replication. Aprataxin protein is highly expressed in the cerebellum, basal ganglia, cerebral cortex, spinal cord and other nervous system tissues [2, 3].

The clinical manifestations among patients with AOA1 are variable regardless of mutation types. The majority of patients present with childhood onset progressive cerebellar ataxia (mean age of onset: 4.3 years; range: 2–10 years), followed by oculomotor apraxia with dis-coordinated eye movement, generalized areflexia and severe peripheral axonal sensorimotor polyneuropathy. Other symptoms including chorea, dystonia, limb weakness, hypoalbuminaemia and hypercholesterolaemiavary among patients. Cognitive impairment has been observed in different degrees among patients although intellect remains normal in some individuals [7, 8].

More than 40 different *APTX* mutations have been identified so far, including nonsense, missense, splice site, frameshift mutations, as well as a complete deletion of the *APTX* gene [2, 3, 8–12]. The most common form of ARCAs in Japan and Portugal is AOA1, with the c.689_690insT (p.Glu232fs) mutation being the most frequent one observed in Japan, while the c.837G>A (p.Trp279X) was the most common mutation among the Portuguese patients [9, 13].

The diagnosis ARCAs is based on the history and the clinical examination, family history, genetic analysis and Sanger sequencing. Clinical examination, including sensorimotor and gait assessment, is required for accurate diagnosis of ARCAs subtypes and magnetic resonance imaging (MRI) is frequently used in assessing the degree of cerebellar atrophy. Currently, no treatment is available for ARCAs in general, and for AOA1 in particular.

The correlation between *APTX* genotype and phenotype has not previously been investigated in Middle Eastern patients, despite the high prevalence of consanguineous marriages in the region. In this study, we recruited two unrelated consanguineous Jordanian families, with affected individuals presenting with undiagnosed movement disorders. Thus, the aim of this study was to identify the possible genetic variants underlying cerebellar ataxia in these two families. In addition, we investigated the phenotypes of the *APTX* mutation identified by whole exome sequencing (WES) and the sensitivity to oxidative DNA damaging agents.

## 2. Materials and methods

### 2.1. Ethics and family recruitment

This study was conducted after receiving ethical approval from the Institutional Review Board (IRB/05/2017), at the Cell Therapy Center/the University of Jordan. Two unrelated consanguineous extended families, family A (n = 18) and family B (n = 4, all were siblings), were recruited into this study (Fig 1). The clinical manifestations varied among patients, but were identical in siblings. Written informed consents were obtained from all participants or their parents for inclusion in the study.

### 2.2. Clinical evaluation and sample collection

Clinical evaluation involved detailed history, neurological examination, full blood count (FBC) and serum chemistry nerve conduction studies (NCS) and Magnetic Resonance Imaging (MRI) (Fig 1A; Table 1). All patients were previously diagnosed with cerebellar atrophy based on the Magnetic Resonance Imaging (MRI) and Nerve Conduction Studies (NCS) results. Brain MRI and NCS were repeated in this study for the probands and some of the affected subjects. Peripheral blood samples for DNA/RNA extraction were collected from 22 study participants, 30 unaffected family members, and ten age and gender matched healthy controls. Full blood count (FBC), alpha-fetoprotein (AFP), albumin, creatine kinase and cholesterol levels were also measured using the same approach. Additionally, skin biopsies for fibroblasts culture were obtained from 6 AOA1 patients and 6 age and gender matched healthy controls.

### 2.3. DNA extraction

Peripheral blood samples for molecular genetic testing were collected from affected members of family A (n = 18) and family B (n = 4) and available parents and unaffected siblings from both families (n = 30). Genomic DNA extraction was carried out using QIAprep Spin Miniprep Kit (Qiagen) according to the manufacturer's instructions. The quantity and quality of the extracted DNA were assessed using NanoDrop 2000 spectrophotometer (ThermoFisher Scientific, USA). DNA was prepared for all patients identified in the pedigree (Fig 1).

### 2.4. Whole exome sequencing and data analysis

Whole exome sequencing was performed for two probands from family A (V: 20, VI: 10) and one proband from family B (IV: 7) by Macrogen service (South Korea). Briefly, exome captures were completed using SureSelect Target Enrichment System (Agilent, Human All Exon Kits v6) (Santa Clara, Californian, U.S.A.) according to the manufacturer's protocol. The libraries were sequenced with Illumina NovaSeq6000 system (Illumina Inc., San Diego, CA). Sequence reads in FASTQ format were aligned to the human reference genome (GRCh37), using the Burrows-Wheeler Aligner (BWA) (http://bio-bwa.sourceforge.net/). Duplicates were removed with Picard (http://picard.Sourceforget.net). WES data were analysed using Genomic Analysis Tool Kit (GATK) (http://www.broadinstitute.org/gatk/) and SnpEff (http://snpeff.sourceforge.

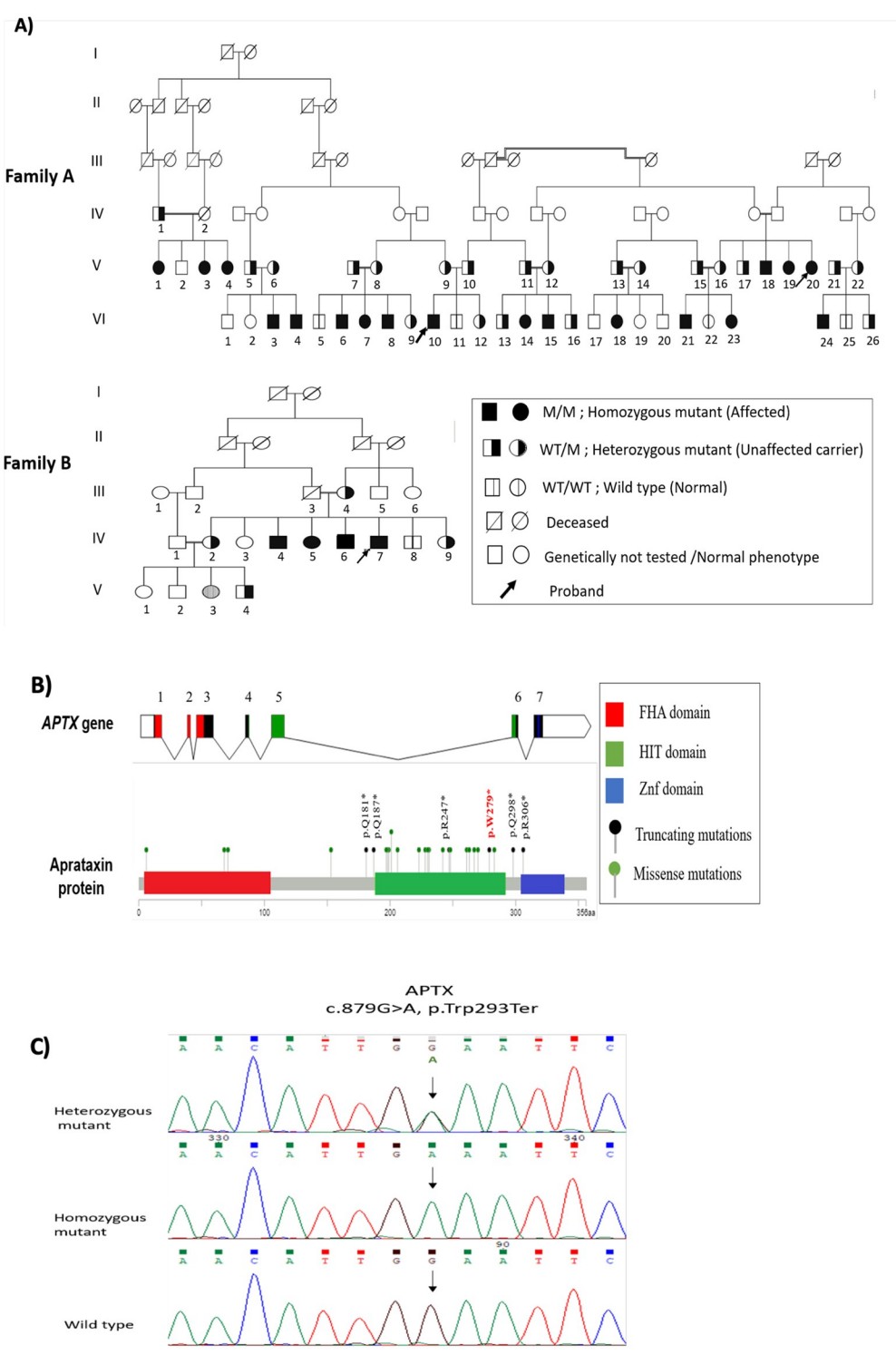

**Fig 1. Identification of disease-causing mutation and segregation analysis. A)** Pedigrees of two unrelated families with AOA1 disease, carrying *APTX* nonsense truncating mutation (c.837G>A; p.W279*). **B)** The location of p.W279* in the *APTX* gene occurs in the histidine triad (HIT) domain of the ataxin protein. **C)** Sanger sequencing of the identified truncating mutation on exon 6 of the *APTX* gene, which occurs as homozygous mutant p.W279* *APTX* gene mutation. The other sequences represent heterozygous mutant (A/G) or wild-type (A/A).

**Table 1. Summary of the major clinical signs for the 22 AOA1 patients in the study.**

| Characteristics | No. of cases/total study subjects | Percentage % |
|---|---|---|
| Number of families | 2 | - |
| Consanguinity | 2/2 | - |
| Age of onset | 3–7 years | - |
| **Clinical Features** | | |
| Peripheral sensory loss | 22/22 | 100.0 |
| Chorea | 22/22 | 100.0 |
| Dysarthria | 22/22 | 100.0 |
| Distal limb muscle wasting and weakness | 22/22 | 100.0 |
| Oculomotor apraxia | 17/22 | 77.3 |
| History of cognitive impairment | 22/22 | 100.0 |
| **Biochemical finding** | | |
| Hypoalbuminaemia | 10/22 | 45.5 |
| Hypercholesterolaemia | 16/22 | 72.7 |
| Increased creatine kinase | 6/22 | 27.3 |
| Increased alpha-fetoprotein | 8/22 | 36.4 |
| **Neurological investigations** | | |
| **Nerve conduction study (NCS)** Peripheral axonal neuropathy | 22/22 | 100.0 |
| **Magnetic resonance imaging (MRI)** Cerebellar atrophy | 22/22 | 100.0 |

net/SnpEff.html). The resulting Variant Call Format (VCF) files were then analyzed using Illumina basespace variant interpreter tool (https://variantinterpreter.informatics.illumina.com/). Variants with read depth less than 10 were excluded. To determine rarity of variants, minor allele frequencies from three publicly available databases were used (1000 Genomes, ExAC, and EVS). Variants occurring with a frequency of $\geq$1% were filtered out. Variant filtering was done under the hypothesis of autosomal recessive inheritance. Then, the variants were filtered using a virtual gene panel of more than 100 neuropathy-related genes. Candidate variants identified through filtering were further analysed for their in silico predicted effect on protein function. The filtered variants were queried in the ClinVar (http://www.ncbi.nlm.nih.gov/clinvar/) and Human Gene Mutation Database (HGMD).

## 2.5. DNA amplification and Sanger sequencing

To identify the presence or absence of the mutation in the *APTX*, the region around the mutation was amplified by PCR and sequenced using the same specific primers through Sanger method, to confirm the segregation of the pathogenic variant detected in patients' families.

Forty-three members from family A (18 affected and 25 healthy) and 9 members from family B (4 affected and 5 healthy) were tested. PCR amplification was carried out using Platinum™ PCR SuperMix (Invitrogen, USA) and 300 nM primer pair flanking the variant sites (ARTX Fwd: 5′–CAGAGGCTTTTCCCATTTTG–′3, ARTX Rev: 5′–AGCTTTCTAGGTCCCCCAAG–′3), 50 ng/µl DNA, and water were mixed to reach a total of 25 µl as the final volume. An initial denaturation step at 94° for 3 minutes was followed by 35 cycles of 94° for 30 seconds, 58° for 30 seconds for annealing, 72° for 30 seconds for elongation, and final extension at 72° for 7 minutes. The PCR products were evaluated using a 2% agarose gel electrophoresis with expected amplicon size of 586 bp. PCR products were then purified by GeneJET PCR purification kit (ThermoFisher, USA) and labelled with BigDye Terminator v3.1 Cycle Sequencing kit (Applied Biosystems, USA). The aforementioned PCR primers (ARTX Fwd and ARTX Rev) were used as sequencing primers. Labelled PCR products were purified using NucleoSEQ® columns

(Macherey-Nagel, Düren, Germany) and then analyzed by ABI 3500 genetic analyzer (Applied-Biosystem). Sequence data were analyzed with SeqA software (Applied Biosystems, USA) and Chromas Pro software (Technolysium LTD, South Brisbane, Australia).

## 2.6. Analysis of mRNA by quantitative reverse-transcription PCR (qRT-PCR)

Extraction of RNA from fibroblasts and peripheral blood samples was done on samples collected from affected individuals and healthy controls subjects with no history of neurodegenerative diseases using TRIzol reagent, and purified with RNeasy Mini kit according to the manufacturer's instructions (Qiagen, USA). First strand cDNA synthesis was performed using GoScript Reverse Transcription System (Qiagen, USA), with 1 μg of RNA from each sample and using both oligo $(dT)^{15}$ and random primers. Each qRT-PCR reaction was composed of GoTaq® qPCR Master Mix (Promega, USA), 300 nM of each forward (5′-CCGACTTCTG GAGAGTGATGA-3′) and reverse (5′-ACGCCCAATCACAACTGCTT-3′) primers specific for *APTX* gene and 20 ng of cDNA template. Mixtures were run on CFX-96 (BioRad) according to the following protocol: 95°C for 3 minutes, 95°C for 10 seconds and 64°C for 1 minute (40 cycles). Samples were performed in technical triplicates, and average Ct-values were used for calculations. Data are represented using the ΔΔCt method. Fold changes were calculated as the average fold change based on GAPDH gene forward primer (5′-CCTGTTCGACAGTCA GCC-3′) and reverse primer (5′-CGACCAAATCCGTTGACTCC-3′), and ß-actin gene forward primer (5′-GGACTTCGAGCAAGAGATGG-3′) and reverse primer (5′-AGCACTGTG TTGGCGTACAG-3′).

## 2.7. Western blot

Western blotting was carried out on whole cell lysates extracted using RIPA buffer (Thermo, USA) on ice. Whole protein was extracted from 4 patients' and 4 healthy controls' fibroblasts and quantified using BCA quantification kit (Abcam, UK). The details were as follows: 20 μg of each sample were denatured in Laemmli buffer for 5 minutes at 95°C. Protein separation was achieved using SDS polyacrylamide gel electrophoresis and transferred onto PVDF membrane using Trans-Blot® SD semi dry transfer system (BioRad, USA). Membranes were blocked with 5% skimmed milk in TBST buffer for 1 hour and then incubated with primary antibodies, diluted in 1% blocking buffer at 4°C overnight. After that, they were washed and incubated with HRP-conjugated secondary antibodies for 1 hour at room temperature. Washed immunoblots were then incubated with TMB substrate (Promega, USA) for 2 minutes. Image acquisition was performed using ChemiDoc imaging system (BioRad, USA). Antibodies used: rabbit anti-aprataxin (1:500; Abcam, ab192598) and mouse anti-β-actin (1:500; Abcam, ab8227), anti-rabbit IgG H&L (1:1000, Abcam, ab6721), and anti-mouse IgG H&L (1:1000, Abcam, 6728).

## 2.8. Cell culture and genotoxicity analysis

Fibroblasts were isolated from skin biopsies of 6 healthy controls and 6 patients carrying the identified *APTX* mutation. Cells were cultured in Advanced DMEM/F-12 (Dulbecco's Modified Eagle Medium/Ham's F-12) (ADMEM, Gibco) containing 10% fetal bovine serum (FBS) (Gibco), 2mM L-glutamine (Life Technologies), 100 U/ml penicillin and streptomycin (Gibco) and maintained in a humidified incubator under 37°C and 5% $CO_2$. Cells were cultured in 96-well plates at a density of 5000 cells/well. The following day, cells were treated in triplicates with increasing concentrations of mitomycin-C (MMC), methyl-methane sulfonate (MMS) and etoposide for 1 hour, and $H_2O_2$ for 30 minutes, under 37°C and 5% $CO_2$. Cells in normal

culture media with no treatment served as a control. To assess the effect of these different genotoxic agents on cell survival, we performed MTT assay [3-(4,5-Dimethylthiazol-2-yl)-2,5-Diphenyltetrazolium Bromide] (Promega, USA) after 48 hours of treatment and the percentage of viable cells was calculated accordingly.

### 2.9. Statistical analysis

The data was analysed using GraphPad Prism software version 6.0. The results were expressed as means ± standard deviation (SD). Statistical significance was analysed by two-way analysis of variance (ANOVA), followed by Bonferroni test for multiple comparisons. A value of $p^* < 0.01$ was considered as significant.

## 3. Results

### 3.1. Clinical presentation of AOA1 patients

All patients had typical symptoms of cerebellar ataxia with disease onset at age of 3–7 years, clinical evidence of cerebellar ataxia, severe sensorimotor peripheral neuropathy with distal muscle wasting and weakness affecting predominantly the lower limbs and chorea (Table 1). All patients were previously diagnosed with Nerve Conduction Study (NCS) and Magnetic resonance imaging (MRI) and they all showed reduced nerve conduction velocities and very small / absent sensory and motor action potentials (S1 Fig). Oculomotor apraxia was observed in most cases of family A (72.2%, n = 13/18) and in all of the four cases of family B (n = 4/4). History of mild to moderate cognitive impairment was noted in 72.7% of all patients. Biochemical analysis showed that serum cholesterol levels were elevated in 16 patients (72.7%), albumin levels were decreased in 10 patients (45.5%), AFP levels were slightly elevated in eight patients (36.4%) and creatine kinase levels were elevated in 6 patients (27.3%, Table 1). Molecular testing of parents and siblings was carried out to confirm variant segregation (Fig 1A–1C). All patients had severe axonal sensorimotor peripheral neuropathy with reduced nerve conduction velocities and very small / absent sensory and motor action potentials and cerebellar atrophy on brain MRI.

### 3.2. Whole exome sequencing and segregation analysis

Genomic DNA was extracted from patients' peripheral blood samples and WES was performed for three probands; two from two different generations of family A (V:20, VI:10) and one from family B (IV:7) (Fig 1A). WES has generated 150 bp average sequence reads length, with 71% on-target reads and 97% target reads coverage at 30X. Based on our filtering criteria, the exome-sequencing analysis of both families resulted in the identification of a homozygous recurrent truncating mutation in *APTX* c.837G>A located at Chr9:32,974,493 and predicted to introduce a stop codon and premature truncation in aprataxin protein (p. W279*) (Fig 1B and 1C). This variant is a nonsense mutation located in exon 6 in the histidine triad (HIT) domain region of the protein, which is predicted to cause complete loss of the aprataxin zinc-finger domain (Fig 1B). The presence of the mutation was also confirmed in all other affected members by Sanger sequencing. *APTX* segregation analysis was performed to identify the mutation among parents and siblings of the affected members. Sanger sequencing confirmed the segregation of *APTX* p.W279* variant and determined that the variant is fully segregated with disease in affected family members, while the healthy subjects were either heterozygous carriers or wild type. All parents tested were heterozygous carriers (Fig 1A and 1C).

### 3.3. *APTX* mRNA and protein analysis

To assess the effect of the detected mutation on gene expression and protein level we used qRT-PCR and western blot. Measurements of mRNA levels using qRT-PCR revealed no significant difference in the level of *APTX* expression between patients and controls in peripheral blood samples when normalized to GAPDH and ß-actin gene levels (Fig 2A). A significant difference in the level of *APTX* was observed in fibroblast samples (**P < 0.001) (Fig 2B). However western blot analysis using anti-aprataxin antibody directed towards the N-terminal region of the protein did not show any detectable levels of the protein. This might suggest that non-sense mediated decay (NMD) could be involved in the clearance of the transcripts with the premature stop codon (Fig 2C).

### 3.4. *APTX* mutation and DNA SSB repair

Since aprataxin protein is involved in SSB system, mutations in this protein may render the *APTX* mutant cells more sensitive to genotoxic agents. In order to investigate sensitivity of patients' cells to genotoxicity, fibroblast cells were exposed to increasing doses of the following DNA-damaging agents: $H_2O_2$, MMC, MMS and etoposide. The number of viable cells was estimated after 48 hours of treatment in drug-free medium. APTX-defective cells were more sensitive to high concentrations of $H_2O_2$ (0.5 mM and 1.0 mM; p<0.010 for both concentrations; ANOVA), MMC (20 μM and 30 μM, p<0.010 and p<0.001, respectively; ANOVA) and etoposide (25 μM and 50 μM, p<0.001 for both concentrations; ANOVA (Fig 3A–3D). No significant changes were observed on any of the MMS concentrations used. Collectively, these results indicate a clear difference between patients and controls, displayed as reduced rates of SSB repair after treating cells with DNA damaging agents.

## 4. Discussion

Aprataxin is a nuclear protein composed of three domains; a forkhead-associated (FHA) N-terminal domain, a histidine triad (HIT) domain in the center and a zinc-finger (ZNF) C-terminus domain [2, 3, 14]. Aprataxin is highly expressed within the cerebellum, basal ganglia, cerebral cortex, and spinal cord [2, 3]. It has been shown that APTX is involved in single-strand DNA repair [15, 16], and mitochondrial transcriptional regulation [17]. It has also been demonstrated that aprataxin resolves the abortive DNA ligation caused by reactive oxygen species and failure of this process results in the accumulation of unrepaired DNA and cellular dysfunction [5]. Studies have confirmed that *APTX* mutations are involved in the degeneration of Purkinjie and cerebellar granular cells which are the most critical pathophysiological features responsible for cerebellar atrophy and ataxia [18, 19].

The *APTX* mutations were first described in Japanese and Portuguese families [2, 3]. To date, several mutations in the *APTX* gene have been reported to cause AOA1 disease. The majority of the previously reported mutations are either frameshift, missense, nonsense, splice-site or deletion mutations [2, 3, 14]. The most common mutation described worldwide is the p.Pro206Leu mutation, while among the European patients is the G837A (p.W279*) mutation [3, 8].

In this study, WES was used to assist in the diagnosis of an unidentified cerebellar ataxia in two unrelated families with a high rate of consanguinity. Currently, WES has emerged as a more straightforward and cost-effective approach to screen pathogenic variants rather than using other traditional molecular diagnostic techniques.

Based on our filtering criteria and the recommendation of the American College of Medical Genetics and Genomics and the Association for Molecular Pathology [20], we identified a recurrent variant of AOA1 in Jordanian families. This *APTX* truncating mutation p.W279*

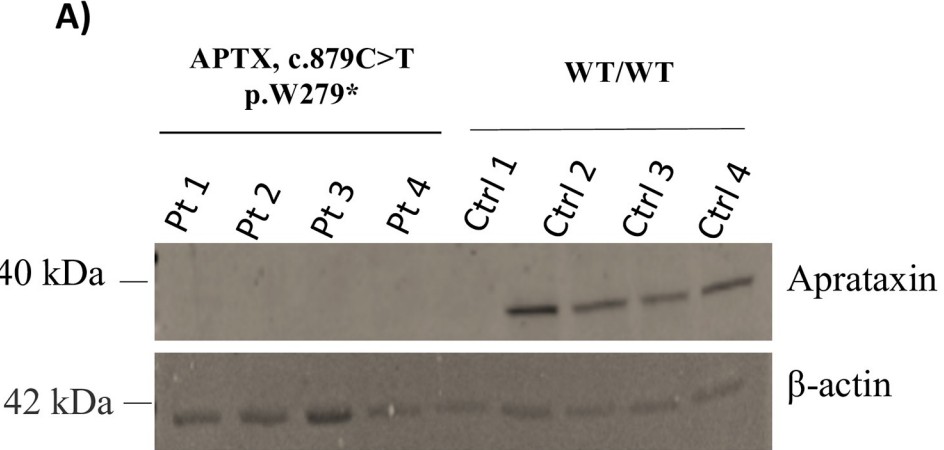

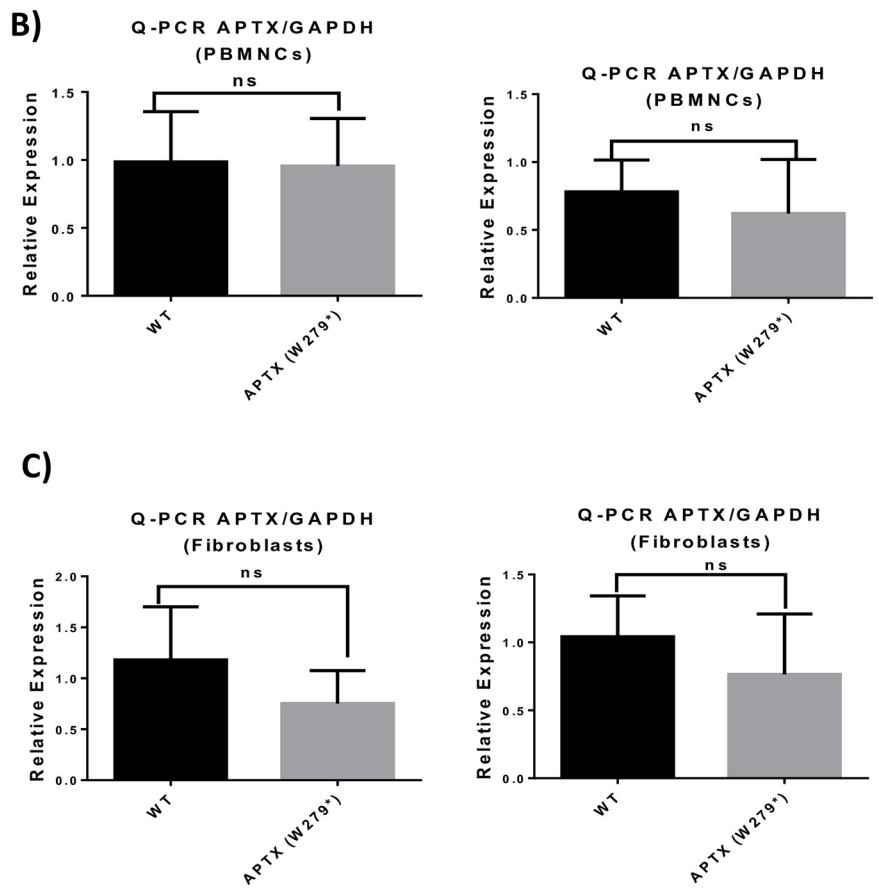

**Fig 2. *APTX* mRNA and protein analysis.** Gene expression level of *APTX* mRNA. Quantitative RT-PCR of *APTX* mRNA in peripheral blood mononuclear cells (A), and in fibroblasts (B), normalized to GAPDH and ß-actin gene levels, respectively. (C) Western blot analysis of WT and mutated APTX proteins. APTX levels were normalized against β-actin protein. Patients carrying *APTX* mutations showed no detectable level of the mutated *APTX*. While healthy controls showed APTX protein with an expected size ~40 kDa. Results represent the means ± standard deviation (SD) calculated from three independent experiments (ns: non-significant, testing was done using two-way ANOVA followed by Bonferroni test).

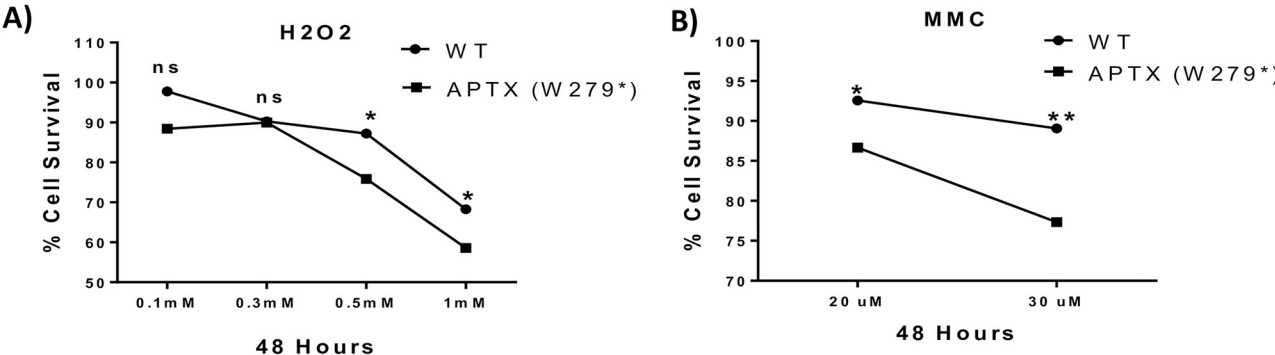

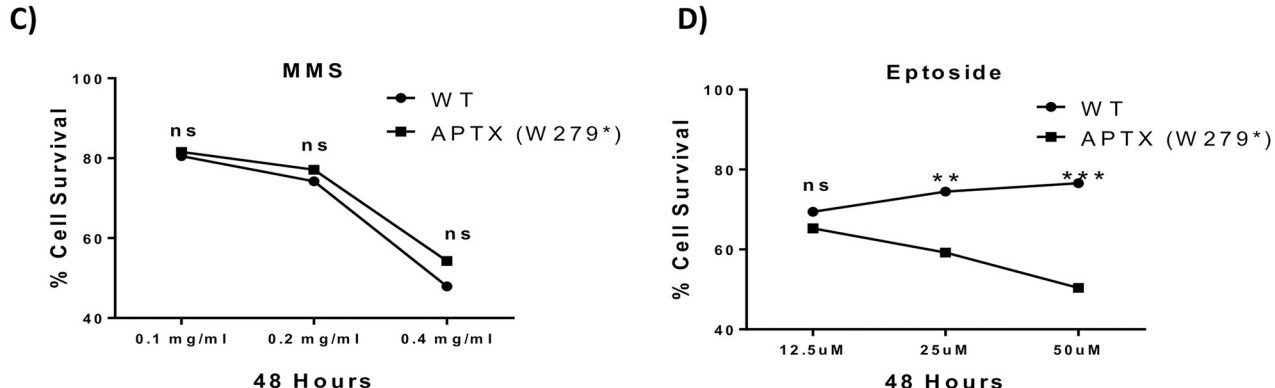

**Fig 3. Sensitivity assay of AOA1 fibroblasts to genotoxins. (A-D)** Percentage of fibroblast cell survival after 48 hours of exposure to increasing concentrations of the following genotoxic agents: $H_2O_2$, MMC, MMS and etoposide at 37˚C, respectively. Results were normalized to untreated controls. A statistically significant difference was observed after treatment of patients' cells with $H_2O_2$, MMC and etoposide. Results represent the means ± SD calculated from three to five independent experiments. (*P < 0.01; **P < 0.001; ns: non-significant).

was found in 22 individuals of 2 unrelated families, family A (n = 18) and family B (n = 4). Early-onset slowly progressive cerebellar ataxia, was found to be the presenting symptom in the majority of our cases described in this study. Oculomotor apraxia was observed in most cases of family A and all four cases from family B. Additionally, all affected members had cerebellar atrophy, severe peripheral neuropathy, distal wasting and weakness of upper and lower limb muscles and chorea.

The identified p.W279* *APTX* mutation in our study has been previously illustrated in the literature [3] (Tranchant et al. 2003; Habeck et al. 2004), however, this is the first report of *APTX* mutation in patients from Jordan. Our identified *APTX* nonsense mutation (p.W279*) is located in exon 6 in the HIT domain region of the protein, which is predicted to cause complete loss of the apataxin zinc-finger domain. Tumbale et.al analysed the effect of different missense and nonsense mutants located within the *APTX* gene on protein solubility, stability and activity and found that: 1) some missense mutations were expressed within the cells, with contrasting solubility depending on the location of the mutation within the protein, 2) two nonsense *APTX* variants (p.R247* and p.W279*) are not expressed in the cells hence they were insoluble and highly destabilized, leading to the loss of the entire Znf domain, 3) the rest of the mutations were expressed within the cells but they were completely insoluble. Thus, they concluded that these substitutions lead to the disruption of the HIT domain structure [21].

In previous studies, the same research group reported that *APTX* truncating mutations are associated with early-onset cerebellar ataxia [8, 9]. The (689insT) truncation mutation in the *APTX* gene revealed the most severe phenotype with a mean age of onset 4.6 years, compared to patients with missense mutations who have less severe phenotype and mean of age of 15.1 years [7].

APTX interferes with the cellular response to genotoxic stress by interacting with DNA repair proteins [22]. The function of the aprataxin protein is to correct the 5'-adenylated DNA resulting from abortive ligations that might occur at any step during DNA replication or oxidative stress repair [5, 23, 24]. Furthermore, APTX protein interacts and forms a complex with XRCC1 and poly-ADP ribose polymerase 1 (PARP-1), implicating this protein with the detection and repair of single-strand DNA breaks [23].

It has been demonstrated that fibroblasts from patients with AOA1 are more sensitive to oxidative damage than normal fibroblasts, and an increase in oxidative DNA damage was observed in the cerebellum of AOA1 patients [23]. To assess the consequences of the identified mutation in our patients' cells, we examined the liability of the patient's fibroblast cells to oxidative stress by exposing them to increasing doses of DNA-damaging agents such as $H_2O_2$, MMS, MMC and etoposide. Thus, APTX-defective cells were significantly more sensitive to high concentrations of $H_2O_2$, MMC and etoposide with no significant changes observed in any of the MMS concentrations used. This might indicate that mutation-carrying cells may have reduced SSBs repair capability following treatment with DNA damaging agents [23]. Such results support the fact that AOA1 pathophysiology is correlated with aprataxin-defective fibroblasts hypersensitivity to reactive oxygen species, which results in an increasing number of SSBs and eventual cell death [15].

Our results are in partial agreement with two similar studies. The first study by Gueven et al reported that cells from AOA1 patients displayed sensitivity to $H_2O_2$ and MMS although similar sensitivity was not reported towards MMC [22]. On the contrary, in the second study reported the lack of toxicity when lymphoblastoid cell line from *APTX* mutated patients treated with $H_2O_2$ compared to control cells [25]. These inconsistencies might be due to the fact that different studies utilize different cell types from patients either affected with the same mutation or different mutations. One other possibility is the level of aprataxin protein varied in patients with certain types of mutations, which might lead to a decreased amount of the protein compared with other types of mutations. Additionally, certain types of mutations might affect protein stability when compared with wild-type aprataxin protein in vitro [15]. The difference in protein stability and amount among these mutated aprataxin proteins might explain the molecular mechanism of phenotype variation.

Also, the number of samples and their mutational background might have played a major role in this variability. Some studies were conducted on one lymphoblast cell line with one single termination mutation, others utilized a group of lymphoblastoid cell lines with different mutational backgrounds including missense as well as termination mutations. Therefore, a more comprehensive study needs to be conducted utilizing a large number of AOA1 patients, grouping them according to their mutational background and utilizing different methods to measure cellular viability to ensure the generation of consistent results.

In summary, this study highlights the importance of genetic diagnosis for neurological diseases such as ataxias, in which the phenotypic characterization is inefficient in defining disease type. Due to the high rate of consanguinity in the Middle East, identifying the genetic variants would help in restricting the spread of hereditary diseases. In the future, prospective screening of the identified mutations would become available for carriers within the same family and it will facilitate the diagnosis of affected children. In Jordan several related families presented with severe clinical manifestations including progressive ataxia, oculomotor apraxia and severe

axonal sensorimotor peripheral neuropathy with no clear diagnosis. Our genetic analysis revealed the *APTX* nonsense variant leading to the termination mutation p.W279*, which resulted in the lack of aprataxin protein expression in patients' fibroblast cells and showed hypersensitivity to several genotoxic agents. There is no available treatment for AOA1 and the current treatment strategies are based on rehabilitation therapy of disabilities. Despite these results, extensive experiments need to be conducted to uncover the exact pathophysiology of such mutations by disease modelling in neuronal cells derived either from autopsies or by cellular reprogramming, which could provide the chance to discover novel therapeutics in the near future.

## 5. Conclusions

The diagnosis of some movement disorders should rely on genetic testing and clinicians should consider this when reporting cases of inherited movement disorders. This study is the first reports of patients withAOA1 in Jordan with p.W279* *APTX* mutation and a heterogenous phenotype. It expands the spectrum of pathogenic *APTX* mutations associated with AOA1 and emphasizes the need of for genetic testing in patients with inherited movement disorders in patients from the Middle East, to assess the spread of founder mutations among consanguineous families.

## Supporting information

**S1 Fig.**
(PDF)

**S1 Raw images.**
(PPTX)

## Acknowledgments

The Authors wish to thank the patients and their families for participation in this study.

## Author Contributions

**Conceptualization:** Nidaa A. Ababneh, Abdalla Awidi.

**Formal analysis:** Nidaa A. Ababneh, Dema Ali, Ban Al-Kurdi.

**Funding acquisition:** Nidaa A. Ababneh.

**Investigation:** Abdee T. Ryalat, Basil Sharrack.

**Methodology:** Nidaa A. Ababneh, Dema Ali, Ban Al-Kurdi, Abdulla M. Alzibdeh, Belal Azab.

**Resources:** Bareqa Salah.

**Supervision:** Abdalla Awidi.

**Writing – original draft:** Nidaa A. Ababneh.

**Writing – review & editing:** Malik Sallam, Abdulla M. Alzibdeh, Basil Sharrack.

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
