## [Decision Letter · Decision Letter 0]

23 Jun 2020

PONE-D-20-17895

Identification of APTX disease-causing mutation in two unrelated Jordanian families with cerebellar ataxia and sensitivity to DNA damaging agents

PLOS ONE

Dear Dr. Ababneh,

Thank you for submitting your manuscript to PLOS ONE. After careful consideration, we feel that it has merit but does not fully meet PLOS ONE’s publication criteria as it currently stands. Therefore, we invite you to submit a revised version of the manuscript that addresses the points raised during the review process.

Although this is only a case report, this is the first case from Jordan of APTX mutations found.  There are some issues that need to be addressed:.

1.  Whole exome sequencing for identification of mutations in one gene seems excessive.  The rationale for this approach should be better described and included in the Discussion.

2.  Address reviewer 1's points regarding the inconsistencies in the patient descriptions from line 229 and Figure1.

3.   Include more information in the Discussion on the impact of the results from this study.

4. Address reviewer 2's comments about treatment of the cell lines.

5.  Ensure that mutation nomenclature is accurate.  (See Reviewer 1's comments).

6.  Ensure that all gene names are italicized.

7.  Include a link to the exome data as all data needs to be available. 

8.  PLOS ONE now requires that submissions reporting blots or gels include original uncropped blot/gel image data as a supplement or in a public repository.   This needs to be added to supplementary material.

9.  As one of the authors is affiliated with Prevention Genetics this needs to be included in the conflicts of interest section. 

10.  Address other points raised by the reviewers.

We look forward to receiving your revised manuscript.

Kind regards,

Amanda Ewart Toland, Ph.D.

Academic Editor

PLOS ONE

Journal Requirements:

"This work was supported by the deanship of scientific research, University of Jordan, grant (2018- 2017/2)"

We note that one or more of the authors are employed by a commercial company: Prevention Genetics.

3.1 Please provide an amended Funding Statement declaring this commercial affiliation, as well as a statement regarding the Role of Funders in your study. If the funding organization did not play a role in the study design, data collection and analysis, decision to publish, or preparation of the manuscript and only provided financial support in the form of authors' salaries and/or research materials, please review your statements relating to the author contributions, and ensure you have specifically and accurately indicated the role(s) that these authors had in your study. You can update author roles in the Author Contributions section of the online submission form.

3.2. Please also provide an updated Competing Interests Statement declaring this commercial affiliation along with any other relevant declarations relating to employment, consultancy, patents, products in development, or marketed products, etc. 

Reviewers' comments:

Reviewer's Responses to Questions

**Comments to the Author**

1. Is the manuscript technically sound, and do the data support the conclusions?

Reviewer #1: Yes

Reviewer #2: Yes

2. Has the statistical analysis been performed appropriately and rigorously? 

Reviewer #1: N/A

Reviewer #2: Yes

3. Have the authors made all data underlying the findings in their manuscript fully available?

Reviewer #1: Yes

Reviewer #2: Yes

4. Is the manuscript presented in an intelligible fashion and written in standard English?

Reviewer #1: Yes

Reviewer #2: Yes

5. Review Comments to the Author

Reviewer #1: Ababneh et al described a case of AOA1. This is the first case of the Jordanian.

Minor comments

L119 FBC should be fully spelled out at first appearance.

L229 All patients had typical symptoms of cerebellar ataxia with disease onset at age of 3-7 years, clinical evidence of cerebellar ataxia, severe sensorimotor peripheral neuropathy with distal muscle wasting and weakness affecting predominantly the lower limbs and chorea (Fig 1A, Table 1) Fig 1A is the familial pedigree tree. The sentence described above does not match the Fig 1A.

L236 albumin levels were elevated in 10 patients (45.5%). In tables 1, the author described as hypoalbuminemia. Usually, in AOA1, serum albumin level is decreased.

L236 AFP levels were observed in eight patients 236(36.4%). Does AFP increase or decreased? In tables 1, the author described as increased AFP. Usually in AOA1, serum AFP level is normal.

L261 APTX p.GW2 79* → APTX p. W279*

Reviewer #2: The authors described a reported mutation, pW279* in Jordan patients with AOA1. Fibroblasts from 6 patients were used.

The paper is well written, but no novel finding is described, except for the first report from Jordan. In addition, there are several points to be addressed.

Major points:

1. The result of MMC, MMS, and H2O2 treatment is not new, with some discrepancies between the present results and the reported ones. The authors described that the discrepancies were attributed to a different mutation. How such mutational differences affect the result is important.

2. Interesting thing is therapy, since the described mutation p.W279* is a common mutation in AOA1. Please describe that prevention of cell death from the genotoxins.

3. If proteasome inhibitors, such as lactascystin, are used, the truncated protein p.W279* would be detectable?

Minor points:

1. The human gene name should be in italicized, ex APTX.

2. Figure legends were not present.

6. PLOS authors have the option to publish the peer review history of their article (what does this mean?). If published, this will include your full peer review and any attached files.

Reviewer #1: No

Reviewer #2: No

---

## [Author Response · Author response to Decision Letter 0]

11 Jul 2020

Response to Editor:

Comment 1. Whole exome sequencing for identification of mutations in one gene seems excessive. The rationale for this approach should be better described and included in the Discussion.

Response: Thank you. The rationale has been described in the discussion and highlighted in yellow. Line: 333

In this study, WES was used to assist in the diagnosis of an unidentified cerebellar ataxia in two unrelated families with high rate of consanguinity. Currently, WES has emerged as a more straightforward and cost-effective approach to screen pathogenic variants rather than using other traditional molecular diagnostic techniques. 

Based on our filtering criteria and the recommendation of the American College of Medical Genetics and Genomics and the Association for Molecular Pathology (20), we identified a recurrent variant of AOA1 in Jordanian families. This APTX truncating mutation p.W279* was found in 22 individuals of 2 unrelated families, family A (n=18) and family B (n=4). Early-onset slowly progressive cerebellar ataxia, was found to be the presenting symptom in the majority of our cases described in this study.

Comment 2. Address reviewer 1's points regarding the inconsistencies in the patient descriptions from line 229 and Figure1.

Response: we have modified the patient descriptions as requested. The reviewer’s points have been addresses and tracked in the same position. 

Comment 3. Include more information in the Discussion on the impact of the results from this study.

Response: Thank you for your suggestion. We have added some data to the discussion as requested. Please see Line: 404 highlighted in yellow.

Comment 4. Address reviewer 2's comments about treatment of the cell lines.

Response: thank you, the comments have been addressed as requested.

Comment 5. Ensure that mutation nomenclature is accurate. (See Reviewer 1's comments).

Response: we have checked the mutation nomenclature and made the required correction.

Comment 6. Ensure that all gene names are italicized.

Response: All gene names have been changed to italic format.

Comment 7. Include a link to the exome data as all data needs to be available. 

Response: we have provided a link at the end of the manuscript; line: 441

Comment 8. PLOS ONE now requires that submissions reporting blots or gels include original uncropped blot/gel image data as a supplement or in a public repository. This needs to be added to supplementary material.

Response: All uncropped blots have been added as supplementary material in PowerPoint document.

Comment 9. As one of the authors is affiliated with Prevention Genetics this needs to be included in the conflicts of interest section. 

Response: Please note that the author ‘Belal Azab’ was affiliated to the University of Jordan during the time of the study. He has recently moved to the Prevention Genetics and there is NO COFLICT of interest to declare. He removed the prevention Genetics affiliation and he only kept the academic affiliations.

Comment 10. Address other points raised by the reviewers.

 Response: Done, please find our point-by-point response in below.

Comment 11: We note that you have provided funding information that is not currently declared in your Funding Statement. However, funding information should not appear in the Acknowledgments section or other areas of your manuscript. We will only publish funding information present in the Funding Statement section of the online submission form.

 Response: Please kindly note that we have removed the funding information from the Acknowledgements sections. Please update our funding statement as the following: This work was supported by the deanship of scientific research, University of Jordan, grant (2018-2017/2). The funding organization did not play a role in the study design, data collection and analysis, decision to publish, or preparation of the manuscript and only provided financial support in the form of research materials.

We note that one or more of the authors are employed by a commercial company: Prevention Genetics.

3.1 Please provide an amended Funding Statement declaring this commercial affiliation, as well as a statement regarding the Role of Funders in your study. If the funding organization did not play a role in the study design, data collection and analysis, decision to publish, or preparation of the manuscript and only provided financial support in the form of authors' salaries and/or research materials, please review your statements relating to the author contributions, and ensure you have specifically and accurately indicated the role(s) that these authors had in your study. You can update author roles in the Author Contributions section of the online submission form.

Response: Please note that the author ‘Belal Azab’ was affiliated to the University of Jordan during the time of the study. He has recently moved to the Prevention Genetics and there is NO COFLICT of interest to declare. He removed the prevention Genetics affiliation and he only kept the academic affiliations.

---

## [Editor Report · Decision Letter 1]

15 Jul 2020

Identification of APTX disease-causing mutation in two unrelated Jordanian families with cerebellar ataxia and sensitivity to DNA damaging agents

PONE-D-20-17895R1

Dear Dr. Ababneh,

We’re pleased to inform you that your manuscript has been judged scientifically suitable for publication and will be formally accepted for publication once it meets all outstanding technical requirements.

Kind regards,

Amanda Ewart Toland, Ph.D.

Academic Editor

PLOS ONE
---

## [Editor Report · Acceptance letter]

22 Jul 2020

PONE-D-20-17895R1 

Identification of APTX disease-causing mutation in two unrelated Jordanian families with cerebellar ataxia and sensitivity to DNA damaging agents 

Dear Dr. Ababneh:

I'm pleased to inform you that your manuscript has been deemed suitable for publication in PLOS ONE. Congratulations! Your manuscript is now with our production department. 

Kind regards, 

on behalf of

Dr. Amanda Ewart Toland 

Academic Editor

PLOS ONE